# Evaluating Genotypes and Seed Treatments to Increase Field Emergence of Low Phytic Acid Soybeans

**Benjamin J. Averitt [1], Gregory E. Welbaum [2], Xiaoying Li [2], Elizabeth Prenger [3], Jun Qin [4] and Bo Zhang [2],\***

1   Department of Crop and Soil Sciences, University of Georgia, Athens, GA 30602, USA; ben.averitt@uga.edu
2   School of Plant and Environmental Sciences, Virginia Tech, Blacksburg, VA 24060, USA;
    welbaum@vt.edu (G.E.W.); xiaoying@vt.edu (X.L.)
3   Department of Plant Science, University of Missouri, Columbia, MO 65211, USA; eprenger@missouri.edu
4   Hebei Academy of Agricultural and Forestry Sciences, Shijiazhuang 050051, China; junqin@soybreeding.com
\*   Correspondence: bozhang@vt.edu

**Abstract:** Low phytic acid (LPA) soybean [*Glycine max* (L.) Merr] genotypes reduce indigestible PA in soybean seeds in order to improve feeding efficiency of mono- and agastric animals, but often exhibit low field emergence, resulting in reduced yield. In this study, four LPA soybean varieties with two different genetic backgrounds were studied to assess their emergence and yield characters under 12 seed treatment combinations including two broad-spectrum, preplant fungicides (i.e., ApronMaxx (mefenoxam: (R,S)-2-[(2,6-dimethylphenyl)-methoxyacetylamino]-propionic acid methyl ester; fludioxonil: 4-(2,2-difluoro-1,3-benzodioxol-4-yl)-1H-pyrrole-3-carbonitrile) and Rancona Summit (ipconazole: 2-[(4-chlorophenyl)methyl]-5-(1-methylethyl)-1-(1H-1,2,4-triazol-1-ylmethyl) cyclopentanol; metalaxyl: N-(methooxyacetyl)-N-(2,6-xylyl)-DL-alaninate)), osmotic priming, and MicroCel-E coating. Two normal-PA (NPA) varieties served as controls. Both irrigated and non-irrigated plots were planted in Blacksburg and Orange, Virginia, USA in 2014 and 2015. Results revealed that three seed treatments (fungicides Rancona Summit and ApronMaxx, as well as Priming + Rancona) significantly improved field emergence by 6.4–11.6% across all genotypes, compared with untreated seeds. Seed priming was negatively associated with emergence across LPA genotypes. Seed treatments did not increase the yield of any genotype. LPA genotypes containing *mips* or *lpa1/lpa2* mutations, produced satisfactory emergence similar to NPA under certain soil and environmental conditions due to the interaction of genotype and environment. Effective seed treatments applied to LPA soybeans along with the successful development of LPA germplasm by soybean breeding programs, will increase use of LPA varieties by commercial soybean growers, ultimately improving animal nutrition while easing environmental impact.

**Keywords:** field emergence; low phytic acid; seed treatment; soybean

## 1. Introduction

Grain soybean [*Glycine max* (L.) Merr] is one of the most important crops for animal feed in the United States due to its high protein content and wide adaptability. Seventy-five percent of phosphorus (P) in soybean seeds is in the form of phytic acid (PA), myo-inositol-1,2,3,4,5,6-hexakisphosphate, which is indigestible for agastric and monogastric animals such as swine, poultry, and most aquatic animals, leading to low feeding efficiency [1]. In addition, other essential minerals, such as calcium, iron, manganese, and zinc, are bound by phytic acid, forming insoluble phytate salts, that render them unavailable, resulting in nutrient deficiencies in monogastric animals [2]. Furthermore,

these nondigestible phytate salts are excreted by animals and become an important source of P pollution detrimental to the environment causing massive algal blooms and fish death [3,4].

Although animal producers have long added synthetic phytase to animal feed to improve PA digestibility, a much more effective method would be the utilization of low-PA (LPA) seeds developed from mutant lines. Three mutant alleles have been reported to create soybean LPA varieties [5]. The first two, *lpa1* and *lpa2*, were both discovered in mutant line CX-1834. These alleles lower phytate by producing a truncated ABC transporter responsible for partitioning PA into seeds [6]. The third mutant allele, *mips1*, is responsible for the first step in PA biosynthesis, catalyzing the NADH-dependent conversion of glucose-6-phosphate to myo-inositol-3-phosphate [7]. However, these mutations not only reduce the seed phytic acid levels in soybean, but also affect the pathways associated with seed development, leading to reduced seed germinability and ultimately low emergence [8,9]. Recent studies showed that many transcriptional genes in biological processes, such as those related to phytic acid metabolism and seed dormancy were involved in this process and the expression diversification of antioxidation-related and hormone-related genes were reported to strongly contribute to variations of emergence rate of LPA soybean lines [8–10]. So far, the mechanism of seed emergence in LPA soybean lines remains unclear and requires further exploration.

Poor field emergence has greatly hindered the use of LPA germplasm in soybean breeding programs [10]. Many attempts have been made to improve emergence in the past few decades. Previous studies showed that soybean seeds produced in temperate environments exhibited higher field emergence than those from tropical/subtropical environments, which illustrates the importance of seed production environment on LPA cultivars for commercial production [11–13]. Maupin and Rainey (2011) reported some seeds derived from the LPA genotype (*mips1*) had field emergence (above 85%) similar to normal-PA (NPA) soybean lines, indicating the potential to develop high emerging LPA soybean lines from natural variations within some LPA mutants [12]. Recently, several new LPA soybean lines (such as 56CX-1273), which display rapid emergence and good agronomic performance, have been developed using traditional crossbreeding methods as well as transgenic technologies [2,14].

Seed treatments improve field emergence of a broad range of crops including soybean. Fungicide treatment is one of the most commonly used to increase soybean stand establishment because it protects seed/seedling from seed- and soil-borne diseases, such as seed rot and damping-off caused by *Phytophthora* spp. [15,16]. Seed priming increases soybean seed vigor, and consequently improves seedling emergence under normal or stressful conditions [17]. Priming involves a controlled hydration procedure followed by redrying applied preplant that allows initial metabolic processes required for seed germination to occur prior to planting resulting in faster germination and uniform field establishment [18]. Additionally, mineral nutrients have been applied preplant as seed coating treatments to improve seedling growth [19,20]. Micro-Cel E, a synthetic calcium silicate, produced by the hydrothermal reaction of diatomaceous silica and high purity lime, can supply plant-essential nutrients and has pesticidal properties. Micro-Cel can be applied as a seed coating and may improve soybean stand establishment.

However, no seed treatment consistently increases the field emergence of LPA soybean lines. The purpose of this study was to apply twelve combinations of four seed treatments: two fungicide treatments (ApronMaxx and Rancona Summit) reported to greatly improve soybean emergence previously [21,22], osmotic priming with potassium phosphate solution and seed coating using Micro-Cel E, to increase field emergence of LPA soybeans. The objective was to: (1) evaluate the seed and seedling vigor of four newly developed LPA soybean varieties (56CX-1283, MD 03-5453, V12-4557, and V12-BB144) with two different genetic backgrounds (i.e., 56CX-1283 and MD 03-5453 having both the *lpa1* and *lpa2* alleles, while V12-4557 and V12-BB144 have the *mips1* allele), and (2) establish a preplant seed enhancement treatment that can effectively improve field emergence and establishment of LPA soybeans.

## 2. Materials and Methods

### 2.1. Plant Materials

Six maturity group V soybean varieties were studied: four LPA and two NPA (Table 1). The four LPA genotypes were 56CX-1283, MD 03-5453, V12-4557 and V12-BB144. 56CX-1283 and MD 03-5453 were developed by the USDA-ARS-Purdue University and University of Maryland, respectively, and contain both the *lpa1* and *lpa2* alleles. V12-4557 and V12-BB144 were developed at Virginia Tech and have the *mips1* allele. Seeds of all six varieties were grown in the same field at the Virginia Tech Kentland Research Farm near Blacksburg, VA using identical agronomic practices the previous year. Seeds of all six genotypes were dried to 9% moisture (dwt basis) after harvest and stored in sealed paper bags maintained in dark in a room maintained at 21 °C until planted the following growing season. The LPA varieties' PA content ranged from 2132 to 4421 ppm. AG 5632 (Bayer, Pittsburgh, PA) and 5002T [23] are both NPA commercial varieties. Their PA content ranged from 5887 to 6116 ppm. MD 03-5453 and V12-4557 have a history of poor field emergence and were not tested in 2014 but were added into the study in 2015.

**Table 1.** The phytic acid (PA) content, genetic source of the low-PA (LPA) trait, and the years planted for each soybean genotype in this study.

| Genotype | LPA Gene | Years Planted | PA Content (ppm) |
|----------|----------|---------------|------------------|
| 5002T | N/A | 2014, 2015 | 6116.10 |
| AG 5632 | N/A | 2014, 2015 | 5886.72 |
| 56CX-1283 | *lpa1/lpa2* | 2014, 2015 | 2486.03 |
| MD 03-5453 | *lpa1/lpa2* | 2015 | 2131.68 |
| V12-4557 | *mips1* | 2015 | 4060.80 |
| V12-BB144 | *mips1* | 2014, 2015 | 4420.50 |

### 2.2. Field Plot Design and Trait Measurement

The experimental design was a triplicated split plot generalized randomized complete block design (GRCBD) wherein the main plots were blocked by the two locations (VT Kentland Farm, Blacksburg and VT Northern Piedmont Research Station, Orange, VA) and split into irrigated and non-irrigated subplots. The Blacksburg location has Hayter loam fine-loamy, mixed, active, mesic Ultic Hapludalfs soil type. The Northern Piedmont site near Orange, VA has Davidson clay loam fine, kaolinitic, thermic Rhodic Kandiudults soil type [24]. Plots were irrigated shortly after planting and kept wet until emergence to simulate damp spring planting conditions typically encountered. All plots were planted using a small-plot mechanical seeder in the last week of May. Orange usually a little warmer than Blacksburg (about 4 °C higher in average in 2014 and 2015) and gets more precipitation (32 more mm in average in 2014 and 2015) than Blacksburg in late May. Each plot was planted in two 3.05 m-long rows spaced 0.82 m apart with 80 seeds per row at a density of 26 seeds per meter. Stand counts were taken at the V1 stage (one set of unfolded trifoliolate leaves) [25]. The plots were once-over destructively harvested in late October (Orange) and early November (Blacksburg). Grain weight and moisture content were recorded for each plot and converted to yield (kg ha$^{-1}$) at 13% moisture on dry weight basis. Phytic acid was measured by high-throughput indirect Fe colorimetry [26].

### 2.3. Seed Treatments

Twelve seed treatment combinations (Table 2) were tested in 2014: MicroCel-E (synthetic calcium silicate, $CaSiO_3$; Manville, Denver, CO); osmotic priming; two fungicides, ApronMaxx (mefenoxam: (R,S)-2-[(2,6-dimethylphenyl)-methoxyacetylamino]-propionic acid methyl ester; fludioxonil: 4-(2,2-difluoro-1,3-benzodioxol-4-yl)-1H-pyrrole-3-carbonitrile; Syngenta Crop Protection, Greensboro, NC) and Rancona Summit (ipconazole:

2-[(4-chlorophenyl)methyl]-5-(1-methylethyl)-1-(1H-1,2,4-triazol-1-ylmethyl) cyclopentanol; metalaxyl: N-(methooxyacetyl)-N-(2,6-xylyl)-DL-alaninate; Valent USA, Walnut Creek, CA, USA); all possible two and three way combinations; and an untreated control. The specific treatments were selected based on prior unpublished results of germination tests. MicroCel-E was ineffective in 2014, so it was excluded in the 2015 trials (Table 2).

**Table 2.** Seed treatments used in this study.

| Treatment | Years Used | Use |
|---|---|---|
| Control | 2014, 2015 | Untreated control |
| ApronMaxx | 2014, 2015 | Broad spectrum fungicide |
| MicroCel-E | 2014 | Weak fertilizer |
| Priming | 2014, 2015 | Post-harvest preplant controlled hydration treatment followed by redrying prior to planting |
| Rancona Summit | 2014, 2015 | Broad spectrum fungicide |
| Priming + Rancona | 2014, 2015 | |
| Priming + ApronMaxx | 2014, 2015 | |
| Priming + MicroCel-E | 2014 | |
| Priming + MicroCel-E + Rancona | 2014 | |
| Priming + MicroCel-E + ApronMaxx | 2014 | |
| MicroCel-E + Rancona | 2014 | |
| MicroCel-E + ApronMaxx | 2014 | |

MicroCel-E was applied to seeds in seed coating. Polyvinyl acetate based-adhesive, $(C_4H_6O_2)_n$ (Elmer's Glue-All, Elmer's Products, Westerville, OH, USA) was diluted 10 times with tap water and misted on seeds (2.5 mL/1000 seeds) in a rotating bowl. Powedered MicroCel-E was slowly added by hand to coat seeds with a thin layer (2.5 mg MicroCel-E/1000 seeds). Seeds were immediately dried at 32 °C in a forced-air dryer for 24 h.

Before seeds were incubated in osmoticum for priming, they were surface sanitized with 30% bleach (8.25%, sodium hypoclorite, NaOCl) solution for 4 min then rinsed in double distilled water (ddH$_2$O). A single layer of seeds was osmotically primed in 3% monopotassium phosphate (KH$_2$PO$_4$) solution in ddH$_2$O on two thicknesses of germination blotter paper (Anchor Paper Co., St. Paul, MN, USA, 9.5 × 9.5 cm) saturated with 20 mL of solution. Seeds were sealed in square clear plastic boxes (10.1 × 10.1 × 3.5 cm OD). Seeds were placed in an incubator at 16 °C for 72 h in dark and force-air dried to their original moisture content.

Seeds were briefly soaked with two aqueous broad-spectrum fungicides, ApronMaxx and Rancona Summit according to label instructions. Twelve mL ApronMaxx was mixed with 10 mL red dye and 78 mL water, and 26 mL Rancona was mixed with 10 mL dye and 64 mL water, respectively. Seeds were treated by applying 2.25 mL of fungicide solution per 1000 seeds in a rotating drum. The seeds were force-air dried to their original moisture content after treated. For treatments with both fungicides and MicroCel-E, solutions were modified to contain either 8 mL Rancona, 7.5 mL dye, and 34.5 mL water; or 6 mL ApronMaxx, 7.5 mL dye, and 36.5 mL water. Three mL of fungicide solutions were applied per 1000 seeds. Once treated, the seeds were dried in a 32 °C dryer for 24 h. All untreated controls were also dried as previously described, so moisture contents of all treatments ranged from 7.5 to 9.5% (dwt basis, 17 h 103 °C).

*2.4. Statistical Analysis*

Correlation analysis of linear lines was calculated using JMP 11 software (SAS Inc, Raleigh, NC, USA) and R software package "corrplot" [27]. A split-split plot analysis of variance was performed using R software with packages "lattice" [28], "car" [29], and "agricolae" [30]. Analysis of variance effects included treatment, genotype, location, irrigation, and replication. The AOV function was used

to perform an ANOVA using the following formula for each year, 2014 and 2015, separately. Means of significant *F*-test were separated by Tukey's Honestly Significant Difference (HSD$_{\alpha\,=\,0.05}$).

## 3. Results

### 3.1. Effects of Genetic and Environmental Factors on Field Emergence

In 2014, an ANOVA revealed significant variation among treatments, varieties, and irrigation regimes. Significant interactions included treatment × genotype, treatment × location, line × location, treatment × irrigation, line × irrigation, treatment × line × location, treatment × location × irrigation, and line × location × irrigation ($p < 0.05$). In 2015, an ANOVA revealed significant variation among treatments and genotype. Significant interactions occurred for treatment × line, treatment × location, line × location, line × irrigation, and treatment × line × irrigation ($p < 0.05$). Field emergence data were averaged separately for each irrigation regime, location in 2014 and 2015 for all soybean genotypes (Table 3). The average field emergence of NPA AG 5632 was 82.6% in 2014, higher than LPA varieties 56CX-1283 (72.1%) and V12-BB144 (70.3%), both of which had significantly higher emergence than NPA genotype 5002T (68.4%). In 2015, 79.5% of 56CX-1283 emerged, followed by NPA genotypes AG-5632 and 5002T without significant differences. For LPA genotype V12-4557, 71.9% of seeds emerged not significantly different from 5002T. Both V12-BB144 (61.8%) and LPA genotype MD 03-5453 (46.7%) were significantly lower than all other genotypes grown in 2015. Across treatments in both years, LPA genotype 56CX-1283 seeds emerged to similar percentages compared to NPA varieties, while the LPA genotype V12-BB144 had variable performance relative to the NPA varieties in Northern Piedmont (Table 3). MD03-5453 and V12-4557 had lower field emergence than the NPA varieties in 2015 with MD03-5453 being the lowest.

For all varieties, field emergence was 4.2% higher in 2014 compared to 2015, probably partly due to the introduction of MD 03-5453 into the study. An overall trend of higher mean emergence in non-irrigated trials compared to irrigated trials occurred across years and locations, indicated that excessive water may have been applied, irrigation increased disease pressure, or some genotypes were sensitive to moist soils. Field emergence percentage varied by location from 77.9% in Orange in 2014 to 68.8% in Blacksburg for 2014. The 2015 mean emergence in Blacksburg was 75.2%, and 63.0% for Orange (Table 3).

### 3.2. General Effects of Seed Treatments on Field Emergence

Seeds treated by Rancona Summit displayed the highest emergence across locations and irrigation regimes in 2014 with an average of 82.1% (Table 4). Rancona Summit was followed in descending order by ApronMaxx (81.9%), the control (80.2%), MicroCel-E + ApronMaxx (78.4%), MicroCel-E + Rancona Summit (76.8%), and Priming + Rancona Summit (76.3%). The untreated control was not significantly different from any seed treatment. Untreated seeds emerged to higher percentages than MicroCel-E, Priming + MicroCel-E + Rancona Summit, Priming, Priming + MicroCel-E, Priming + MicroCel-E + ApronMaxx, and Priming + ApronMaxx.

Emergence data were collected on fewer treatments in 2015 after ineffective treatments were identified in 2014. Untreated seed emergence was 67.4%, significantly lower compared with the three most effective seed treatments. Rancona Summit (79.0%) and ApronMaxx (76.6%) producing the highest emergence (Table 4). The Priming + Rancona Summit treatment performed better compared to the control in 2015 with emergence of 73.8%. Untreated control emergence was significantly higher than Priming and Priming + ApronMaxx treatments.

**Table 3.** Field emergence of two normal-PA (NPA) and four LPA soybean varieties grown at Blacksburg and Orange in 2014 and 2015.

| Line | Irrigation | Phytate | Emergence % | | | | | |
|------|-----------|---------|-------------|-----------|-------------|---------|---------|---------|
| | | | 2014 Only | 2015 Only | 2014 BB [1] | 2014 O | 2015 BB | 2015 O |
| **5002T** | I [2] | Normal | 64.6 * | 71.7 * | 65.4d | 71.4c | 76.5bc | 74.7cd |
| | N | | 72.1 | 79.5 | | | | |
| | Mean | | 68.4c [3] | 75.6ab | | | | |
| **AG-5632** | I | Normal | 79.2 * | 77.4 | 76.8b | 88.4a | 85.5a | 72.2cd |
| | N | | 86.0 | 79.9 | | | | |
| | Mean | | 82.6a | 78.7a | | | | |
| **56CX-1283** | I | *lpa1/lpa2* | 69.7 * | 79.2 | 67.6d | 76.5b | 82.4ab | 76.7bc |
| | N | | 74.4 | 79.9 | | | | |
| | Mean | | 72.1b | 79.5a | | | | |
| **V12-BB144** | I | *mips1* | 66.0 * | 56.5 * | 65.2d | 75.3b | 77.1bc | 46.8f |
| | N | | 74.6 | 66.9 | | | | |
| | Mean | | 70.3b | 61.8c | | | | |
| **MD 03-5453** | I | *lpa1/lpa2* | - | 44.8 | - | - | 54.0e | 39.6g |
| | N | | - | 48.7 | | | | |
| | Mean | | - | 46.7d | | | | |
| **V12-4557** | I | *mips1* | - | 68.1 * | - | - | 75.8bc | 68.2d |
| | N | | - | 76.0 | | | | |
| | Mean | | - | 71.9b | | | | |
| | Mean | | 73.3 | 69.1 | 68.8 | 77.9 | 75.2 | 63.0 |

[1] BB = Blacksburg; O = Orange. [2] I = Irrigated; N = Non-irrigated. [3] Values within a column with values for a single year (2014 only; 2015 only or within a set of columns with values for a single year (2014 BB and O; 2015 BB and O) followed by the same letter are not significantly different based on Tukey's HSD ($\alpha$ = 0.05). * Indicates a significant difference ($p \leq 0.05$) in emergence between irrigation regimes for the corresponding genotype.

**Table 4.** Average field emergence and Tukey's separation of means for 12 seed treatment combinations in 2014 and 2015.

| Treatment | Irrigation | Emergence % | | | | | |
|---|---|---|---|---|---|---|---|
| | | 2014 Only | 2015 Only | 2014 BB [1] | 2014 O | 2015 BB | 2015 O |
| **Control (untreated)** | I [2] | 77.8 | 64.4 | 74.0fghij | 86.3a | 71.0b | 63.8cd |
| | N | 82.6 | 70.3 | | | | |
| | Mean | 80.2ab [4] | 67.4c | | | | |
| **ApronMaxx** | I | 77.9 * | 73.7 | 78.3cdefg | 85.5ab | 83.4a | 69.8bc |
| | N | 85.8 | 79.5 | | | | |
| | Mean | 81.9a | 76.6ab | | | | |
| **Rancona** | I | 78.6 * | 76.2 | 76.8defgh | 87.4a | 86.3a | 72.0b |
| | N | 85.6 | 81.7 | | | | |
| | Mean | 82.1a | 79.0a | | | | |
| **Microcel** | I | 72.2 | - | 69.7ijk | 79.2bcdef | - | - |
| | N | 76.7 | - | | | | |
| | Mean | 74.5c | - | | | | |
| **Priming** | I | 63.7 * | 55.7 | 63.4lmn | 73.3fghijk | 62.6d | 52.0e |
| | N | 73.0 | 58.8 | | | | |
| | Mean | 68.4de | 57.3d | | | | |
| **MxA [3]** | I | 76.6 | - | 74.9efghi | 81.8abcd | - | - |
| | N | 80.2 | - | | | | |
| | Mean | 78.4abc | - | | | | |
| **MxR** | I | 73.0 * | - | 70.2ijk | 83.5abc | - | - |
| | N | 80.7 | - | | | | |
| | Mean | 76.8bc | - | | | | |
| **PA** | I | 61.0 | 58.0 | 56.9o | 68.0jklm | 68.9bcd | 52.6e |
| | N | 63.9 | 63.5 | | | | |
| | Mean | 62.4f | 60.7d | | | | |
| **PR** | I | 71.5 * | 70.3 * | 71.1hijk | 81.2abcde | 79.5a | 68.5bcd |
| | N | 81.3 | 77.4 | | | | |
| | Mean | 76.3bc | 73.8b | | | | |
| **PM** | I | 60.2 * | - | 62.2mno | 68.9ijkl | - | - |
| | N | 70.9 | - | | | | |
| | Mean | 65.5ef | - | | | | |
| **PxMxA** | I | 59.3 * | - | 59.8no | 67.6klm | - | - |
| | N | 68.1 | - | | | | |
| | Mean | 63.7f | - | | | | |
| **PxMxR** | I | 66.8 | - | 67.9jklm | 71.9ghijk | - | - |
| | N | 73.0 | - | | | | |
| | Mean | 69.9d | - | | | | |
| | Mean | 73.3 | 69.1 | 68.8 | 77.9 | 75.3 | 63.1 |

[1] BB = Blacksburg; O = Orange; [2] I = Irrigated; N = Non-irrigated; [3] M = MicroCel-E; A = ApronMaxx; R = Rancona; P = Priming; [4] Values within a column with values for a single year (2014 only; 2015 only or within a set of columns with values for a single year (2014 BB & O; 2015 BB & O) followed by the same letter are not significantly different based on Tukey's HSD ($\alpha$ = 0.05). * Indicates a significant difference ($p \leq 0.05$) in emergence between irrigation regimes for the corresponding.

### 3.3. Effects of Seed Treatments on Field Emergence by PA Phenotype

Analysis of the two NPA and four LPA soybean varieties and six treatments showed patterns of field emergence among phytic acid types in 2015 (Figure 1). The untreated control treatment average field emergence for the two *mips1* genotypes was 70.5%, while untreated seed emergence for the two *lpa1/lpa2* genotypes was 59.3%. Seeds treated with Rancona Summit, ApronMaxx,

and Priming + Rancona had slightly higher average field emergence compared to untreated seeds. Priming and Priming + ApronMaxx tended to decrease emergence relative to untreated seeds. The overall mean of untreated emergence of the LPA varieties was 64.0%, while emergence of the two NPA varieties was 72.2%. This confirms that the LPA varieties may have lower inherent emergence than NPA varieties, although this could be largely due to the inclusion of the low-emerging genotype MD 03-5453 and the likelihood of greater disease incidence in wet soils. ApronMaxx and Rancona fungicide treatments, however, improved the emergence of LPA varieties by between 12.9% and 14.1%. This improvement suggests that fungicide treatments have the potential, when optimized, to improve LPA seed emergence to essentially the same percentages as NPA varieties.

*3.4. Effect of Seed Treatments on Field Emergence of LPA and NPA Genotypes*

The application of some seed treatments to the six different soybean LPA and NPA genotypes significantly improved field emergence, but effects were specific to each line (Table 5). The treatment × genotype combinations produced significant variations in both 2014 and 2015. In 2014, no treatment × genotype combination significantly improved emergence over untreated seeds of the same genotype. ApronMaxx, Rancona Summit, MicoCel-E, MicroCel-E + ApronMaxx, MicroCel-E + Rancona Summit, and Priming + Rancona Summit treatments increased emergence relative to untreated controls for at least one of four genotypes grown in 2014 (Table 5). For LPA genotypes, nearly all treatments decreased emergence relative to the control by 2–37%. The exceptions to this decrease was line V12-BB144, which ApronMaxx, Rancona Summit, and MicroCel-E + ApronMaxx treatments slightly increased (0.2–1.4%) or had no effect on emergence. MicroCel-E, Priming, Priming + ApronMaxx, Priming + MicroCel-E, Priming + MicroCel-E + ApronMaxx, and Priming + MicroCel-E + Rancona Summit each had a significant decrease in emergence relative to the control for at least one of four genotypes (Table 5).

In 2015, several treatment × genotype combinations improved emergence compared to untreated seeds of the same genotype. ApronMaxx and Rancona Summit treatments increased emergence over untreated seeds of LPA line MD 03-5453, and the highest emergence was 69.5% (Table 5). Rancona Summit as well as Priming + Rancona Summit increased emergence for NPA line 5002T compared to the control. ApronMaxx, Rancona Summit, and Priming + Rancona Summit treatments increased emergence for most genotypes relative to untreated, but the increases were not significant except as described previously. Both Priming and Priming + ApronMaxx treatments decreased emergence for five out of six genotypes from 0.7–27% compared to the control (Table 5).

*3.5. Effect of Treatments, Genotypes, and Treatment × Line Interactions on Yield*

No seed treatment significantly increased yield of any genotype in either year (Table 6). However, significant differences in yield existed among genotypes in both years. In 2014, NPA line AG-5632 had the highest mean yield across all treatments at 5145 kg ha$^{-1}$, followed by LPA line 56CX-1283 (4849 kg ha$^{-1}$), NPA line 5002T (4768 kg ha$^{-1}$), and V12-BB144 (4479 kg ha$^{-1}$). Only 56CX-1283 and 5002T were not significantly different. In 2015, the yield of 56CX-1283 and AG-5632 were not significantly different. Genotypes 5002T, V12-BB144, and V12-4557 yielded significantly less than either 56CX-1283 or AG 5632. The lowest yielding line was MD 03-5453 at 1211 kg ha$^{-1}$, significantly lower than all other genotypes. No seed treatment increased yield compared to the control for each respective genotype. Seed treatment differences existed among genotypes, but not for treatments applied to the same genotype.

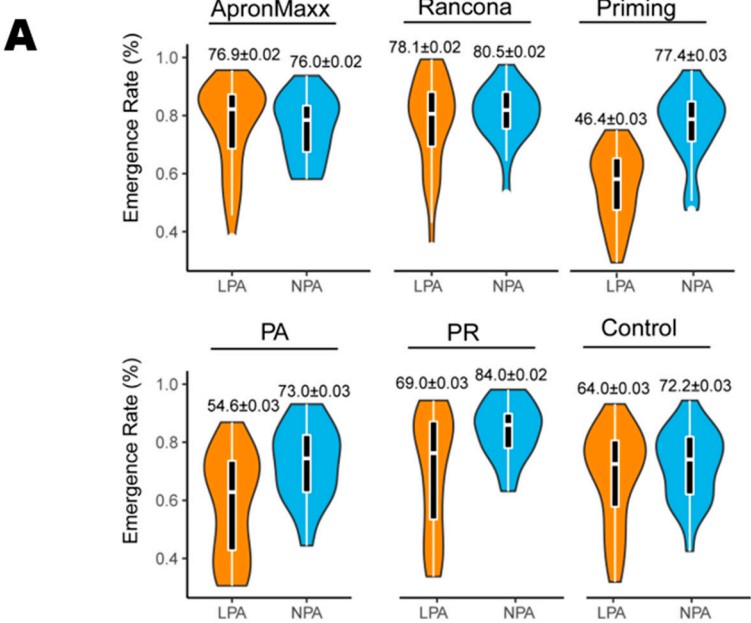

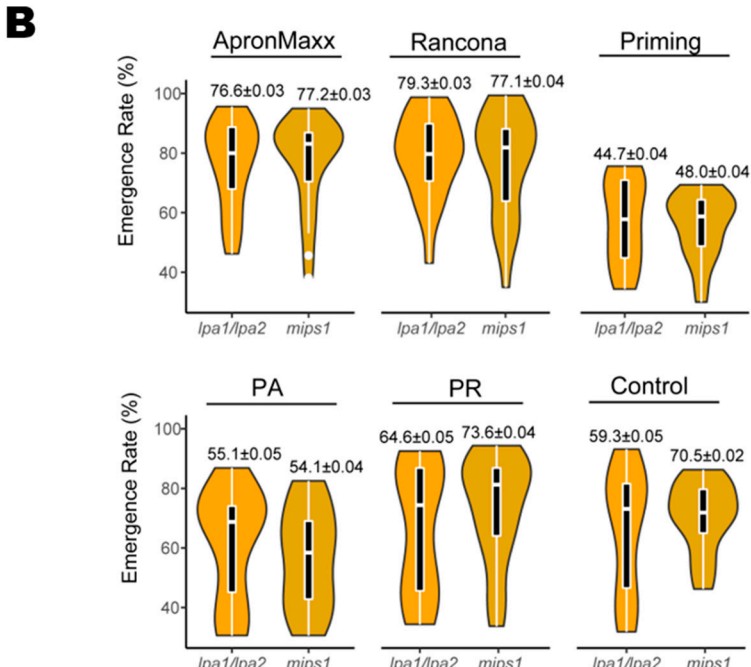

**Figure 1.** Field emergence for six seed treatments across different soybean varieties grown in 2015. Emergence rates of four low phytic acid (LPA) and two normal phytic acid (NPA) soybean varieties (**A**) as well as two LPA varieties with *lpa1* and *lpa2* alleles and two other LPA varieties with the *mips1* allele (**B**) were calculated. White horizontal lines at the center of each box show median values. The bounds of each black box show the quartiles, and the upper and lower bars show the maximum and minimum values, respectively. This image was drawn using ggplot2 package. Numbers above the violin plot indicate the means ± standard deviations. PA: Priming + Rancona; PR: Priming + ApronMaxx.

**Table 5.** Effects of seed treatments on field emergence in NPA and LPA soybeans and Tukey's separation of means in 2014 and 2015.

| | | 2014 | | | | | | | | | | | | |
|---|---|---|---|---|---|---|---|---|---|---|---|---|---|---|
| | | Emergence % | | | | | | | | | | | | |
| Line | Phytate | C [1] | A | R | M | P | MxA | MxR | PA | PR | PM | PxMxA | PxMxR | Mean |
| **5002T** | NPA [2] | 77.6 | 84.4 | 81.3 | 70.4 | 66.0 * | 72.1 | 75 | 53.3 * | 70.7 | 51.5 * | 50.8 * | 67.1 * | 68.4c [3] |
| **AG-5632** | NPA | 81.8 | 87.3 | 87.3 | 82 | 79.4 | 87.7 | 82.5 | 80.3 | 84.5 | 80 | 77.6 | 80.8 | 82.6a |
| **56CX-1283** | LPA | 85.3 | 79.9 | 83.6 | 73.9 * | 69.4 * | 76.4 | 77.9 | 48.4 * | 79.1 | 71.0 * | 58.4 * | 61.3 * | 72.1b |
| **V12-BB144** | LPA | 75.9 | 75.9 | 76.1 | 71.6 | 58.5 * | 77.3 | 72 | 67.7 | 70.2 | 59.6 * | 68 | 70.5 | 70.3b |
| **Mean** | - | 80.2ab | 81.9a | 82.1a | 74.5c | 68.4de | 78.4abc | 76.8bc | 62.4f | 76.3bc | 65.5ef | 63.7f | 69.9d | - |
| | | 2015 | | | | | | | | | | | | |
| **5002T** | NPA | 65.1 | 77.8 | 78.8 * | | 76.3 | | | 71.6 | 84.0 * | | | | 75.6ab |
| **AG-5632** | NPA | 79.3 | 74.3 | 82.3 | | 78.6 | | | 74.3 | 84.1 | | | | 78.7a |
| **56CX-1283** | LPA | 80.4 | 85.4 | 88.6 | | 63.2 * | | | 74.6 | 85.1 | | | | 79.5a |
| **V12-BB144** | LPA | 67.9 | 72.5 | 71.6 | | 41.3 * | | | 47.9 * | 69 | | | | 61.8c |
| **MD 03-5453** | LPA | 36.3 | 67.7 * | 69.5 * | | 26.2 | | | 35.5 | 44.1 | | | | 46.7d |
| **V12-4557** | LPA | 72.9 | 81.8 | 82.6 | | 56.7 * | | | 60.4 | 78.6 | | | | 71.9b |
| **Mean** | - | 67.4c | 76.6ab | 79.0a | | 57.3d | | | 60.7d | 73.8b | | | | - |

[1] C = control; A = ApronMaxx; R = Rancona Summit; M = MicroCel-E; P = Priming; [2] NPA = normal phytic acid; LPA = low phytic acid; [3] Values followed by the same letter within bordered columns or rows are not significantly different based on Tukey's HSD ($\alpha$ = 0.05). * Indicates a treatment is significantly different from the control treatment for the corresponding genotype.

**Table 6.** Effects of seed treatments on yield of six soybean lines grown at Blacksburg and Orange and Tukey's separation of means in 2014 and 2015.

| Line | Phytate | C[1] | A | R | M | P | MxA | MxR | PA | PR | PM | PxMxA | PxMxR | Mean |
|---|---|---|---|---|---|---|---|---|---|---|---|---|---|---|
| | | | | | | | **2014** Yield kg ha$^{-1}$ | | | | | | | |
| **5002T** | NPA[2] | 4782 | 4983 | 4909 | 4782 | 4459 | 4842 | 5057 | 4815 | 4701 | 4519 | 4573 | 4768 | 4768b[3] |
| **AG-5632** | NPA | 4936 | 5151 | 5151 | 5091 | 4956 | 5077 | 5387 | 5185 | 5219 | 5185 | 5151 | 5205 | 5144a |
| **56CX-1283** | LPA | 4882 | 4882 | 4922 | 4829 | 4882 | 4936 | 5010 | 4546 | 5098 | 4808 | 4707 | 4721 | 4849b |
| **V12-BB144** | LPA | 4304 | 4431 | 4566 | 4734 | 4465 | 4425 | 4492 | 4580 | 4526 | 4156 | 4539 | 4499 | 4479c |
| **Mean** | - | 4728ab | 4862ab | 4889ab | 4856ab | 4687ab | 4822ab | 4990a | 4782ab | 4896ab | 4667b | 4741ab | 4795ab | - |
| | | | | | | | **2015** | | | | | | | |
| **5002T** | NPA | 3477 | 3618 | 3551 | | 3558 | | | 3564 | 3511 | | | | 3544b |
| **AG-5632** | NPA | 4284 | 3827 | 3867 | | 4270 | | | 3867 | 3901 | | | | 4008a |
| **56CX-1283** | LPA | 4001 | 3921 | 4304 | | 4479 | | | 4102 | 4055 | | | | 4143a |
| **V12-BB144** | LPA | 3685 | 3894 | 3732 | | 3141 | | | 3268 | 3470 | | | | 3531b |
| **MD 03-5453** | LPA | 1110 | 1506 | 1076 | | 659 | | | 1446 | 1439 | | | | 1211c |
| **V12-4557** | LPA | 3571 | 3800 | 3531 | | 3477 | | | 3443 | 3289 | | | | 3517b |
| **Mean** | - | 3416a | 3430a | 3336a | | 3302a | | | 3309a | 3255a | | | | - |

[1] C = control; A = ApronMaxx; R = Rancona Summit; M = MicroCel-E; P = Priming [2] NPA = normal phytic acid; LPA = low phytic acid [3] Values followed by the same letter within bordered columns or rows are not significantly different based on Tukey's HSD ($\alpha$ = 0.05).

## 4. Discussion

A major use of grain soybean is animal feed because of its high protein content. However, high levels of PA in soybean seeds may lead to animal mineral and protein malnutrition. In addition, phytic acid phosphorus excreted by monogastric animals such as poultry, swine, and fish can become a pollutant. These problems have provided plant geneticists with an incentive to develop LPA soybean varieties [31]. However, PA is also important for the growth and development of soybean seedlings because it is a primary storage reserve of phosphate in seeds. Phosphate is an essential component of adenosine triphosphate (ATP) that provides energy necessary for seedling growth and development. Thus, phosphorus is essential for the general health and vigor of developing seedlings. Reducing seed phytate by re-engineering synthesis pathway often has the unintended consequence of reducing seedling vigor and harming crop establishment [32]. Unfortunately, LPA soybeans often exhibit lower field emergence, making them problematic to grow particularly during stressful growing conditions.

This study included MD 03-5453 and 56CX-1283 expressing *lpa1/lpa2* homologs responsible for a low phytic acid phenotype (Table 1). In combination, *lpa1/lpa2* lower the PA content to about 25% of NPA genotypes while the remaining 75% phosphorus is inorganic [1,33]. This study also included V12-4557 and V12-BB144 genotypes, expressing *mips1*, another allele responsible for LPA soybeans (Table 1). Compared with *lpa* mutants, *mips1* mutants have higher seed PA content where it usually accounts for 50% of total phosphorus. However, *mips1* mutants increase feed efficiency for mono-and a-gastric animals with the added benefit of a modified, beneficial sugar profile. Since they have higher PA than *lpa* mutants, germination would be predicted to be similar to wild-type soybeans.

The emergence data for LPA genotypes were inconsistent between genotypes and years compared to NPA genotypes. The *mips1* LPA genotype V12-BB144 showed higher emergence than NPA 5002T in 2014 but significantly lower emergence in 2015, while emergence of *mips1* LPA genotype V12-4557 was essentially the same as 5002T in 2015, the only year it was grown. LPA genotype MD 03-5453, containing *lpa1/lpa2*, emerged to lower percentages than all others in 2015. However, except for MD 03-5453 and V12-BB144 in 2015, the other two LPA genotypes exhibited average field emergence of around 70% or greater. This suggests that LPA genotypes containing *mips* or *lpa1/lpa2* mutations, can produce satisfactory emergence if seeds are carefully produced and stored properly prior to planting. Other studies have shown that LPA genotypes, *lpa1/lpa2* as well as *lpa1/lpa2* with GmIPK2 silenced, produced satisfactory germination or field emergence [14,34]. Maupin and Rainey (2011) reported average emergence of between 74–84% for varieties with *mips* or *lpa1/lpa2* mutations tested across 12 unique environments [12]. Anderson and Fehr (2008) reported up to 81.0% field emergence for *lpa1/lpa2* mutants from various seed sources [11].

Final grain yield was only loosely correlated with field emergence. Grain yield was not significantly affected by seed treatments. Soybean plants compensate by producing more pods per plant at wider spacings, so when emergence is slightly reduced, as was the case for most treatments in this study, yield was not affected [35].

Inconsistencies in emergence data among genotypes, treatments, and years were influenced by several important seed quality factors irrespective of the genetic-controlling phytate accumulation. This study was conducted at a cooler location at 650 m elevation (Blacksburg) and a warmer climate (Orange) at lower elevation in the Virginia Piedmont with vastly different soil types. Edaphic differences between locations such as soil microbes, soil texture, water holding capacity, etc., likely contributed to variation in emergence among genotypes and treatments complicating the conclusions about the role of seed phytate on emergence.

Environmental factors regulating seed fill can negatively impact seed vigor expression when seeds are grown for propagation. High temperatures, for example, during seed development decreased seed weight, caused shriveling, and decreased seed quality of soybean [36] and reduced soybean seed vigor in the absence of mechanical injury and seedborne diseases [37]. Drought stress on the parent soybean plant had little effect on seed quality although yields were reduced [38,39]. To mitigate

maternal environmental effects on quality, seeds of all six genotypes used in this study were produced in the same season and location.

Seed vigor, another important determinant of emergence particularly under stressful field conditions, is affected by a number of factors such as: seed maturity at harvest, physical seed damage during harvest and transport, and improper storage. McDonald (1985) reviewed losses in seed vigor from maturation to planting in soybean as well as identifying seed quality tests that detected physical seed damage [40]. Although the seeds tested in the current study were grown at the same location to minimize differences in seed vigor, tailoring the time of harvest for highest seed vigor was not a focus. Delayed harvest may reduce soybean seed vigor [41]. Maximum seed quality and vigor often correlates with maximum dry weight accumulation [42]. However, physiological maturity can be better detected morphologically in some seeds. For example, maximum seed dry weight was not the best indicator of physiological maturity in common bean as pod color change [43]. Bean seeds with low quality produced fewer nodules, less nodule weight, and less nitrogen fixation that resulted in less plant growth and yield [44]. Vigor tests are more sensitive measures of seed quality than the standard germination tests or field stand counts, which are often used to assess germination of low phytate genotypes. Vigor tests in future studies could yield additional valuable information about the poor emergence sometimes observed in LPA soybean genotypes.

Improper post-harvest handling compromises seed quality. Open storage in combination with high relative humidity and high temperature can quickly result in a loss of seed vigor. All genotypes were grown and stored under identical conditions, so differences were most likely due to seed genotypes and not environment. Chauhan (1985) found the growing points of the embryonic axis in soybean were most prone to aging than other seed tissues [45]. This illustrates that seed tissues do not age simultaneously, and cotyledons may be healthy even after embryonic axis is damaged resulting in poor emergence. In this study all seeds were adjusted to the same moisture content after harvest and stored in paper bags at a room temperature. Seeds may have aged under these conditions, but all genotypes were exposed to the same aging conditions.

Hoy and Gamble (1985, 1987) found that soybean seed size had no effect on specific growth rate or seedling weight from planted seeds possessing no mechanical injury [46,47]. No improvement in speed of field emergence or final yield was detected when soybean seeds were separated into varying seed density classes [48]. Thus, in this study, seeds were not sized before field planting due to the poor correlation between seed size and seed vigor.

Seed treatments may benefit field emergence and were investigated as a strategy for improving establishment of LPA soybean genotypes. While there is no consensus about the exact reason for low emergence by LPA soybean genotypes, there are likely causes. Because some fungicide treatments improved LPA emergence, disease pressure before emergence is likely higher for LPA than NPA genotypes. Soil-borne pathogens are possibly the main cause of poor emergence in some seed lots planted in wet soils. Cellular leakage occurs in all seeds during imbibition because of cell membrane damage that occurs during desiccation that must be repaired. Cells repair membrane damage during hydration, and the duration of this process depends on seed quality. Electrolyte leakage is widely used vigor test to assess soybean seed quality [49]. Aged seeds leak more solutes and electrolytes than newly harvested undamaged high vigor seeds. Evidence suggests that some LPA genotypes naturally leak more compounds that attract seed/seedling pathogens because 75% of their phosphorus is inorganic [1,33]. The loss of inorganic P from the cytoplasm of LPA varieties due to imbibitional leakage could increase disease since leaked P can attract soil-borne pathogens to the emerging seedling explaining the benefits of fungicide treatments [50]. In addition, Douglass, et al. (1993) found a negative correlation between seed sugar content of differing sweet corn genotypes and emergence in cold soils [51] while this correlation is still unclear for soybean.

The lower emergence in irrigated plots supports the hypothesis that LPA are more prone to fungal attack since moist soils would create favorable conditions for disease development possibly leading to greater seed/seedling mortality. Fungicide was the most effective seed treatment in this

study. Both fungicides significantly increased the field emergence of LPA genotype MD 03-5453, supported the hypothesis higher pre-emergence disease pressure could be a major cause of the low field emergence of LPA soybeans.

Osmotic priming is a common preplant controlled hydration treatment often applied to high value flower and vegetable seeds. Benefits of priming include faster germination, advancing seed maturity, leaching of inhibitors, and removal of dormancy. However, priming also reduces the storage life of seeds [18]. Priming treatments are less often applied to lower value agronomic seeds because the cost of application may outweigh benefits. Osmotic priming was used to increase germination rate so that seedlings would establish before diseases could infect vulnerable young plants [52]. Surprisingly, osmotic priming did not improve establishment and reduced field emergence similar to hydroprimed soybeans [53]. In the current study, seeds were primed in potassium phosphate solution which was not removed by washing at the end of treatment. These salts combined with the leakage of electrolytes that occurred during the controlled hydration priming treatment, described above, likely increased susceptibility to pathogenic attack as nutrients surrounded seeds and aided the proliferation of plant pathogens. Similarly, MicroCel-E, a calcium silicate processed from diatomaceous earth with a low salt index that contains small amounts of plant nutrients including phosphate, was applied as a seed coating. Ideally the nutrients would stimulate early seedling growth and the antipathogenic properties of diatomaceous earth may provide protection from insect and fungal predation. However, MicroCel-E consistently failed to improve emergence unless it was combined with a fungicide. The antifungal properties of diatomaceous earth were likely ineffective against seedling pathogens and insect predation. The fertilizer may have attracted and stimulated microbial growth unless fungicides Rancona Summit or ApronMaxx were present.

Emergence results were variable in this study, making it difficult to draw simple conclusions about treatments or genotypes. This is because of the complexity of factors interacting to affect field emergence. Edaphic stressors in the field commonly reduce emergence compared to results obtained from standardized laboratory germination tests conducted under near ideal conditions. In some plots, LPA seeds with *lpa1/lpa2* and *mips1* alleles had satisfactory field emergence compared to NPA. In other trials emergence of LPA genotypes was less than NPA likely because of conditions favoring seedling disease due to greater metabolite leakage from LPA seeds because of the altered phosphate and sugar metabolism which increased mortality. Osmotic priming and diatomaceous earth coating were ineffective. In some plots, seed fungicide treatments improved emergence of certain genotypes likely by protecting seeds/seedlings from pathogens that reduce emergence.

**Author Contributions:** Conceptualization, B.J.A., G.E.W. and B.Z.; methodology, G.E.W. and B.Z.; software, B.J.A., E.P. and J.Q.; formal analysis, B.J.A., E.P.; investigation, B.J.A.; resources, B.Z.; data curation, B.J.A.; writing—original draft preparation, B.J.A.; writing—review and editing, G.E.W., X.L., E.P. and J.Q., and B.Z.; visualization, B.J.A.; supervision, B.Z.; project administration, G.E.W. and B.Z.; funding acquisition, B.Z. All authors have read and agreed to the published version of the manuscript.

**Funding:** This research was funded by United Soybean Board.

**Acknowledgments:** This work was conducted under US multi-state project, W-468. Thanks to Luciana Rosso, Tom Pridgen, Steve Gulick, and Andy Jensen for technical support. Thanks to Hwasoo Shin for helping to make the graph.

**Conflicts of Interest:** The authors declare no conflict of interest.

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
