# Peer review of "Evaluating Genotypes and Seed Treatments to Increase Field Emergence of Low Phytic Acid Soybeans"

_agriculture, doi:10.3390/agriculture10110516_

Round 1
Reviewer 1 Report
Comments are included in the attached manuscript

Author Response
Reviewer #1
Comment 1: Line 49: Delete “a” either "declined" or consider "reduced. Line 94: Use “grown” instead of “growing”.
Response 1: Thank you for your suggestions. We have changed the words as you suggested in lines 50 and 96.
Comment 2: Line 106: Since differences associated with field conditions are mentioned in the Discussion as possible reasons for the variance in results some data should be provided in terms of climatic conditions, soil physical properties, type, time, and amount of irrigation. Anything more in this direction would be helpful.
Response 2: Thank you for bringing this to our attention. We have added the requested information to lines 110-120.
Comment 3: Line 172: that is true only in O.
Response 3: We totally agree with you, and we added “in Northern Piedmont” that is Orange in line 186.
Comment 4: Line 173: They are lower but not as much as in the case of MD and not statistically different from 5002T.
Response 4: Thank you for the comment. We have revised the line 186-187: “MD03-5453 and V12-4557 had lower field emergence than the NPA varieties in 2015 with MD03-5453 being the lowest”.
Comment 5: Line 177: how much was that? How much more than the recommended amount of water? In this case soil physical properties should be mentioned.
Response 5: Thank you for the comment. We have added the description of irrigated plots to line 115-117. “Plots were irrigated shortly after planting and kept wet until emergence to simulate damp spring planting conditions typically encountered”. We also described soil types in line 110 to 115.
Comment 6: Table 5: The range of values is extremely high. In this case a histogram will be helpful to display the sample size, standard deviation, or error.
Response 6: Thank you for the comment. We have added a figure (Figure 1) to display values better on page 9.
Comment 7: Line 228 “The application of seed treatments to the six different soybean LPA and NPA genotypes significantly improved field emergence” Is this true?
Response 7: Yes, Line 253-254: “The application of some seed treatments to the six different soybean LPA and NPA genotypes significantly improved field emergence, but effects were specific to each line”.
Comment 8: Line 231-233: This is confusing. Consider instead: However, some treatments increased emergence relative to untreated controls
Response 8: Thank you for the comment. We added “some” to lines 253, same as above.
Comment 9: Line 252: As mentioned above the range of values is high. Did you check the distribution of values in the different populations (treatments) before comparing mean values? What was the sample size?
Response 9: No, we didn’t compare the distribution/replication of each genotype across three replications at one location (if this is what you meant). We just calculated the mean as mean indicates the average emergence. The sample size is 160 seeds per plot as one replication with three replications for each genotype under each treatment, which is described in line 121.
Comment 10: Line 259: Add “the” before “yield”. Line 261: Change “Both..and..” with “either.. or.”
Response 10: We have changed the words as you suggested in lines 285 and 287.
Comment 11: Line 287: Do you mean higher than lpa mutants? it is not clear? Line 289: Compared to lpa mutants?
Response 11: Yes, we have edited the sentence of line 313 as “Compared with lpa mutants, mips1 mutants have higher seed PA content where it usually accounts for 50% of total 287 phosphorus”. We also edited the sentence of line 316 as “Since they have higher PA than lpa mutants, the germination would be predicted to be similar to wild-type soybeans.”
Comment 12: Line 311: This was not clear in the Materials and Methods. What consisted a "vast" difference? Such differences should be mentioned.
Response 12: We have added the soil info. to line 110-115.
Comment 13: Line 353-354: Add a reference.
Response 13: This is our assumption. We have revised line 380-384: “Because some fungicide treatments improved LPA emergence, disease pressure before emergence is likely higher for LPA than NPA genotypes. Soil-borne pathogens are possibly the main cause of poor emergence in some seed lots planted in wet soils.” We have searched the literatures, but didn’t find any references to have the same conclusion.
Comment 14: Line 369: This is speculation- it is better to say: " create favourable conditions for disease development"
Response 14: Thank you for the suggestion. We have rewritten the sentence as “The lower emergence in irrigated plots supports the hypothesis that LPA are more prone to fungal attack since moist soils would create favorable conditions for disease development possibly leading to greater seed/seedling mortality” in line 399-400.
Comment 15: Line 372: could be? Otherwise it should be proven experimentally. Line 389: Add “a” before “fungicide”. Line 393: Delete “many”.
Response 15: We have changed “is” with “could be” in line 402, add “a” in line 420, and delete “many” in line 425.

Reviewer 2 Report
It is a well written paper and presents detailed results of a comprehensive study examining the potential benefit of seed treatments, including both fungicidal protectants and osmotic primers, for ameliorating the poor viability and reduced emergence observed when soybean varieties with low or reduced phytic acid are are planted. The experiments were conducted with adequate replication and appropriate well organized experimental design appropriate for examining a number of possible factors influencing the performance of the varieties.
Author Response
Reviewer #2
Comment 1: It is a well written paper and presents detailed results of a comprehensive study examining the potential benefit of seed treatments, including both fungicidal protectants and osmotic primers, for ameliorating the poor viability and reduced emergence observed when soybean varieties with low or reduced phytic acid are planted. The experiments were conducted with adequate replication and appropriate well-organized experimental design appropriate for examining a number of possible factors influencing the performance of the varieties.
Response 1: Thank you for the comments!
Reviewer 3 Report
This is interesting article about some LPA soybean varieties and their emergence and yield characters under seed treatment combinations of two broad-spectrum, preplant fungicides, osmotic priming and MicroCel-E coating.
The abstract presents very clear the objectives of the study and the main results obtained. The introduction is supported by well selected bibliographic data. All bibliographic sources are fairly recent and correctly mentioned in text.
Although emergence results were very variable and the conclusions about treatments or genotypes are not simple, the study can be useful to researchers in the field of soybean breeding. However, in my opinion, because the field emergence can be affect by many complexities of factors, I suggest to the authors to mention in the Materials and Methods section which were the agronomic practices used for growing of all six soybean varieties.
Some minor observations are the following:
Line 33. To avoid any confusion, please specify in parentheses the meaning of the P symbol (phosphorus).
Line 132. ″ddH20″. I suppose it's about Double distilled water. Please specify this in parentheses.
Author Response
Reviewer #3
Comment 1: This is interesting article about some LPA soybean varieties and their emergence and yield characters under seed treatment combinations of two broad-spectrum, preplant fungicides, osmotic priming and MicroCel-E coating. The abstract presents very clear the objectives of the study and the main results obtained. The introduction is supported by well selected bibliographic data. All bibliographic sources are fairly recent and correctly mentioned in text.
Although emergence results were very variable and the conclusions about treatments or genotypes are not simple, the study can be useful to researchers in the field of soybean breeding. However, in my opinion, because the field emergence can be affected by many complexities of factors, I suggest to the authors to mention in the Materials and Methods section which were the agronomic practices used for growing of all six soybean varieties.
Response 1: Thank you for your suggestion. We have added the detailed information of the agronomic practices used in this study in the Materials and Methods section (please see line 110-120 in the revised manuscript).
Comment 2: Line 33. To avoid any confusion, please specify in parentheses the meaning of the P symbol (phosphorus).
Response 2: Thank you for bringing this to our attention. We revised line 33-34 “Seventy-five percent of phosphorus (P) in soybean seeds is in the form of phytic acid (PA)”.
Comment 3: Line 132. ″ddH20″. I suppose it's about Double distilled water. Please specify this in parentheses.
Response 3: Yes, it is double distilled water. Thank you for your suggestion. We have clarified it in line 145.